# Clinical Implications of Krüpple-like Transcription Factor KLF-14 and Certain Micro-RNA (miR-27a, miR-196a2, miR-423) Gene Variations as a Risk Factor in the Genetic Predisposition to PCOS

**DOI:** 10.3390/jpm12040586

**Published:** 2022-04-06

**Authors:** Rashid Mir, Nizar H. Saeedi, Mohammed M. Jalal, Malik A. Altayar, Jameel Barnawi, Abdullah Hamadi, Faris J. Tayeb, Sanad E. Alshammari, Nabil Mtiraoui, Mohammed Eltigani M. Ali, Faisel M. Abuduhier, Mohammad Fahad Ullah

**Affiliations:** 1Faculty of Applied Medical Science, University of Tabuk, Tabuk 71491, Saudi Arabia; jbarnawi@ut.edu.sa (J.B.); fabu-duhier@ut.edu.sa (F.M.A.); 2Department of Medical Laboratory Technology, Faculty of Applied Medical Science, University of Tabuk, Tabuk 71491, Saudi Arabia; nsaeedi@ut.edu.sa (N.H.S.); mjalal@ut.edu.sa (M.M.J.); maltayar@ut.edu.sa (M.A.A.); a.aldhafri@ut.edu.sa (A.H.); f.tayeb@ut.edu.sa (F.J.T.); 3Department of Pharmacology & Toxicology, Faculty of Pharmacy, University of Hail, Hail 55476, Saudi Arabia; sanad-pharm@hotmail.com; 4Laboratory of Human Genome and Multifactorial Diseases, Faculty of Pharmacy, University of Monastir, Monastir 5000, Tunisia; mtiraouinabil@yahoo.fr; 5King Salman Military Hospital, Tabuk 47512, Saudi Arabia; tigoalgarbawi@yahoo.com

**Keywords:** polycystic ovary syndrome, hyperandrogenism, polymorphic gene variation, disease risk, biochemical characterization, endocrine profile

## Abstract

Polycystic ovary syndrome (PCOS) is a disorder with a symptomatic manifestation of an array of metabolic and endocrine impairments. PCOS has a relatively high prevalence rate among young women of reproductive age and is a risk factor for some severe metabolic diseases such as T2DM, insulin insensitivity, and obesity, while the most dominant endocrine malfunction is an excess of testosterone showing hyperandrogenism and hirsutism. MicroRNAs have been implicated as mediators of metabolic diseases including obesity and insulin resistance, as these can regulate multiple cellular pathways such as insulin signaling and adipogenesis. Genome-wide association studies during the last few years have also linked the Krüpple-like family of transcription factors such as KLF14, which contribute in mechanisms of mammalian gene regulation, with certain altered metabolic traits and risk of atherosclerosis and type-2 DM. This study has characterized the biochemical and endocrine parameters in PCOS patients with a comprehensive serum profiling in comparison to healthy controls and further examined the influence of allelic variations for miRNAs 27a (rs895819 A > G), 196a2 (rs11614913 C > T), 423 (rs6505162C > A), and transcription factor KLF14 (rs972283 A > G) gene polymorphism on the risk and susceptibility to PCOS. The experimental protocol included amplification refractory mutation-specific (ARMS)-PCR to detect and determine the presence of these polymorphic variants in the study subjects. The results in this case–control study showed that most of the serum biomarkers, both biochemical and endocrine, that were analyzed in the study demonstrated statistically significant alterations in PCOS patients, including lipids (LDL, HDL, cholesterol), T2DM markers (fasting glucose, free insulin, HOMA-IR), and hormones (FSH, LH, testosterone, and progesterone). The distribution of Krüppel-like factor 14 rs972283 G > A, miR-27a rs895819 A > G, and miR-196a-2 rs11614913 C > T genotypes analyzed within PCOS patients and healthy controls in the considered population was significant (*p* < 0.05), except for miR-423 rs6505162 C > A genotypes (*p* > 0.05). The study found that in the codominant model, KLF14-AA was strongly associated with greater PCOS susceptibility (OR 2.35, 95% CI = 1.128 to 4.893, *p* < 0.022), miR-27a-GA was linked to an enhanced PCOS susceptibility (OR 2.06, 95% CI = 1.165 to 3.650, *p* < 0.012), and miR-196a-CT was associated with higher PCOS susceptibility (OR 2.06, 95% CI = 1.191 to 3.58, *p* < 0.009). Moreover, allele A of KLF-14 and allele T of miR-196a2 were strongly associated with PCOS susceptibility in the considered population.

## 1. Introduction

PCOS is a complex disorder associated with an array of endocrine and metabolic impairments (Ali 2015). The disorder is gender-specific, with a relatively high prevalence rate among young women of reproductive age, often leading to infertility [1]. PCOS is also a risk factor for some severe metabolic diseases such as T2DM, insulin insensitivity, and obesity, while the most dominant endocrine malfunction is an excess of testosterone, causing hyperandrogenism and hirsutism [2]. The prevalence of PCOS ranges from 5 to 13% among young women, whereas within these patients, the prevalence of infertility varies from 70 to 80% [3]. Some representative features of metabolic syndrome in PCOS in both obese and non-obese patients include insulin resistance and high serum insulin levels, which are considered to interfere in ovulation and promote synthesis of ovarian testosterone [4]. Furthermore, patients with this disorder are presented with an enhanced risk of fetus congenital heart and neural tube inadequacy, preeclampsia, miscarriages, and preterm births, which are associated with maternal metabolic and endocrine systemic abnormalities [5,6]. The human genome is a common site for polymorphic variations, particularly single-nucleotide polymorphisms (SNPs), which appear as an alternative base replacement in a DNA sequence (genes or regulatory regions). Sometimes such variations may be functional and contribute to the risk of a disease [7]. Certain allelic polymorphisms demonstrate genetic profiles that are predictive of disease severity and advanced disease progression [8]. There is emerging evidence that link microRNAs as mediators of metabolic diseases including obesity and insulin resistance, as these can regulate multiple cellular pathways such as insulin signaling and adipogenesis [9]. MicroRNAs are single-stranded RNAs, which are noncoding and contain about ≈21 nucleotides, with a function to regulate the target gene expression by mRNA decay and inhibition of the transcript translation. The human genome has ≈45,000 estimated miRNA-targeting sites that are considered significant in regulating the expression of about ≈60% of genes [10]. The presence of certain SNPs in the miRNA gene has been known to influence its expression, maturation, or ability to bind the mRNA target site to form RNA-induced silencing complex [11]. Such polymorphic gene variations observed in miRNAs have been linked with the pathophysiology of several metabolic diseases including cardiovascular disorders and diabetes [12,13]. Moreover, altered follicular fluid, granulosa cells, serum, plasma, and tissue profiles related to a number of miRNAs have been recorded in patients with PCOS, and it is believed that these might contribute as facund mediators in the etiology and pathology of the disease [14]. Recently, significant associations of miRNA-196, miRNA-423, and miRNA 27a have been reported with the risk and susceptibility of certain diseases [15,16,17,18,19]. Genome-wide association studies during the last few years have linked the Krüpple-like family of transcription factors such as KLF14, which contribute to the mechanisms of mammalian gene regulation, with certain altered metabolic traits and risk of atherosclerosis and type-2 DM [20,21]. KLF14 is known to be an imprinted gene wherein only the maternal-inherited allele is functional, and interestingly, the gene has been linked to the regulatory processes of placental development and embryogenesis [22,23].

The present work from our laboratory represents some of the findings from a multiphase study on polycystic ovary syndrome, seeking to identify and analyze the significance of certain gene loci with polymorphic variations, which may be contribute to the disease susceptibility and clinical outcomes. Herein, we present data that examine the association of miRNAs 27a (rs895819 A > G), 196a2 (rs11614913 C > T), 423 (rs6505162C > A), and transcription factor KLF14 (rs972283 A > G) polymorphic gene variations with the risk and susceptibility to PCOS. Moreover, biochemical characterization of clinical variables has also been described, showing certain complex effects of the disease in patients as compared to the healthy controls.

## 2. Methodology

### 2.1. Study Participants and Criteria

PCOS has a complex etiology, and as per the practice, the disease is diagnosed if the patient shows any two of the following three conditions: ovulatory dysfunction, androgen excess, and multiple ovarian cysts. Complexity of the disorder requires the co-assessment of clinical, endocrine, and ultrasonography results. The PCOS cases were confirmed using protocols established as per the 2003 Rotterdam criteria [24]. The population subjects in the study included Saudi Arabs, while non-Arab Saudi or expatriates were excluded from the study. The study recruited 230 subjects at the outpatient department of the Obstetrics/Gynecology Unit of King Salman Military Hospital-Tabuk, Saudi Arabia, which included 115 clinically confirmed PCOS patients and 115 gender-matched controls. 

#### 2.1.1. Biochemical Serum Profile

The study examined the biochemical serum profiles of the study subjects including fasting glucose, insulin level, HbA1c, serum lipids, and hormones in the first phase of the study. Fasting glucose level was determined with a hexokinase kit (Cobas Integra 800; Roche, Munich, Germany). An ELISA kit (DRG-EIA) was used to measure the total insulin as per the vendor’s specification. A HOMA calculator (www.dtu.ox.ac.uk/homa/index, (accessed on 25 January 2022)) performed the computation of the HOMA-IR index. ELISA kits standardized for each assessment [25] were used to record the serum levels of different hormones including estradiol, testosterone, TSH, FSH, LH, and progesterone. Colorimetric determination (Integra 800; Roche) provided the total cholesterol, triglycerides, LDL, and HDL in serum.

#### 2.1.2. Extraction and Qualitative Assessment of Genomic DNA

Our laboratory used the DNA extraction kit (Cat # 69506/Qiagen, Hilden, Germany) for the extraction of genomic DNA from the peripheral blood samples of the patients and healthy controls, following the vendor’s protocol. Nuclease-free water was used to finally solubilize the isolated DNA that was then stored at 4 °C for further use. The DNA was quantitated with a NanoDrop™ (Thermo Scientific, Waltham, MA, USA), and qualitative assessment of the extracted DNA was carried optically as a ratio of A_260nm_/A_280nm_ (1.83–1.99).

#### 2.1.3. Genotyping of KLF-14, miR-27, miR-423, and miR-196a2 

Amplification-refractory mutation system PCR (ARMS-PCR) was optimized using tetra-primers that were specific for KLF-14 rs972283, miR-27 rs895819, miR-423-rs6505162, and miR-196a2-rs11614913 polymorphisms. ARMS primers are shown in Table 1.

The PCR reaction was performed in a total reaction volume of 12 µL that contained template DNA (50 ng), FO −0.12 µL, RO −0.12 µL, FI −0.12 µL, RI −0.12 µL (25 pmol of desired primer), and 6 µL Green PCR Master Mix (2X) (Cat# M712C/Promega, Madison, WI, USA). The nuclease-free ddH_2_O was used to adjust the final reaction volume to 12 µL in each PCR tube. The primers used in the study have been described previously [26,27,28,29]. The thermocycling conditions used were as follows: initial denaturation at 95 °C for 9 min followed by 38 cycles of denaturation at 95 °C for 36 s; annealing for 40 s at 60 °C for KLF-14 rs972283 genotyping, 63 °C for miR-27 rs895819 genotyping, 62 °C for miR-423-rs6505162 genotyping, and 61 °C miR-196a2-rs11614913 genotyping; extension at 72 °C for 45 s, followed by the final extension at 72 °C for 12 min. 

### 2.2. Gel Electrophoresis and PCR Product Visualization

PCR-amplified products were resolved through the agarose gel electrophoresis (2%) and stained with SYBR safe dye. The gel image was visualized on a gel documentation system from Bio-Rad, Hercules, CA, USA.

### 2.3. KLF-14 rs972283 

Outer primers F1 and R1 amplify the outer region of the KLF14, generating a band of 437 bp that acts as a control for DNA purity. Primers F1 and R2 amplify the A allele, generating a band of 221 bp, and primers F2 and R1 generate a band of 274 bp from the G allele. 

### 2.4. MiR-27a rs895819 

Outer primers FO and RO amplify the outer region of the miR-27, generating a band of 353 bp that acts as a control for DNA purity. Primers FI and R2 amplify the A allele, generating a band of 226 bp, and primers F2 and R1 generate a band of 184 bp from the G allele. 

### 2.5. MiR-423 rs6505162

Outer primers F1 and R1 amplify the outer region of the miR-423, generating a band of 336 bp that acts as a control for DNA purity. Primers FI and R2 amplify the C allele, generating a band of 160 bp, and primers F2 and R1 generate a band of 228 bp from the A allele. 

### 2.6. MiR-196a2 rs11614913 

Primers F1 and R1 flank the exon of the miR-196a2, generating a band of 297 bp that acts as a control for DNA purity. Primers F1 and R2 amplify the C allele, generating a band of 153 bp, and primers F2 and R1 generate a band of 199 bp from the T allele. 

## 3. Statistical Analysis 

Statistical analysis was performed for the assessment of the comparative data for the PCOS patients and healthy controls using the SPSS 16.0 software package (Chicago, IL, USA). Comparison of KLF-14 rs972283, miR-27 rs895819, miR-423-rs6505162, and miR-196a2-rs11614913 genotyping frequency and biochemical characteristics was carried out by chi-squared analysis and Fisher’s exact test. Further, the evaluation of the Hardy–Weinberg equilibrium was performed by a χ^2^ test to compare the observed genotype frequencies within the case–control subjects. In all the observations, the *p*-value was deemed significant when it was less than 0.05. Multivariate analysis examined the association between KLF-14 rs972283, miR-27 rs895819, miR-423-rs6505162, and miR-196a2-rs11614913 genotypes and susceptibility to PCOS by comparing the odds ratios (ORs), risk ratios (RRs), and risk differences (RDs) with 95% confidence intervals (CIs) [30].

## 4. Results

### 4.1. Comparative Biochemical Profiling Showed Altered Clinical Markers in PCOS Patients

Since PCOS is a complex disorder, its effects are observed in a number of clinical parameters that are significantly altered in patients. As reported in Table 2, most of the tested biomarkers in patients showed a marked difference when compared to healthy controls. In patients, the fasting glucose and insulin were higher, as these PCOS patients developed T2DM and insulin resistance during the course of the disease. Lipid profile and BMI of the patients group also related to the metabolic impairments, which may lead to obesity in such patients. In the endocrine assessment, it was observed that testosterone levels were higher in patients, which represents hyperandrogenism as one of the key features of the disorder. Further, the serum progesterone was also significantly altered in the patient group compared to the healthy subjects, whereas differences in estradiol and prolactin levels were not significant. Clinical features in PCOS are thus diverse and demonstrate the accumulation of various detrimental outcomes, which affect the well-being of such patients and their quality of life.

### 4.2. HWE Equilibrium Showed no Deviation

The distributions of genotype and allele frequencies of the SNPs located in the Krüppel-like factor 14 rs972283 C > T, miR-27a rs895819 A > G, miR-196a-2 rs11614913 C > T, and miR-423 rs6505162 C > A presented no deviation in the HWE model (all *p* > 0.05) in the control group. On the basis of this, 10% samples from the normal control group were randomly selected to review genotyping results, indicating that the accuracy rate was more than 99%.

### 4.3. Allele Distribution and Genotype Frequency in PCOS Patients and Controls (p-Values) for Krüppel-like Factor 14 rs972283 G > A Genotypes 

In PCOS cases, the genotype frequencies for GG, GA, and AA were 37.38%, 32.71%, and 29.90%, respectively, and in healthy controls, GG, GA, and AA genotype frequencies were 40.86%, 45.21%, and 13.91%, respectively (Table 3). The distribution of Krüppel-like factor 14 rs972283 G > A genotypes between PCOS patients and healthy controls was found to be significant (*p* < 0.011). Furthermore, our result shows that the frequency of allele A (fA) was significantly higher among PCOS patients than in healthy controls (0.46 vs. 0.36).

### 4.4. Association between Krüppel-like Factor 14 rs972283 G > A Genotypes and PCOS Susceptibility as Determined by Multivariate Analysis

Statistical analyses based on logistic regression references such as odds ratio (OD) and risk ratio (RR) with 95% confidence intervals (CI) were performed by multivariate analysis for each group to estimate the association between Krüppel-like factor 14 rs972283 G > A genotypes and risk to PCOS; the data are reported in Table 4. As is shown, our results demonstrated that in the codominant model, the KLF14- AA genotype was strongly associated with greater PCOS susceptibility, with OR 2.35, 95% CI = 1.1286 to 4.8932, RR = 1.62 (1.0390 to 2.5280), *p* < 0.022. Similarly, these results also show that in the recessive inheritance model, the KLF14-AA vs. KLF14-(GG + GA) genotype was strongly associated with enhanced PCOS susceptibility, with OR 2.64, 95% CI = 1.134 to 5.16, RR = 1.70 (1.12 to 2.59), *p* < 0.004. Moreover, in comparative allelic distribution and risk assessment, it was found that the allele A was strongly associated with PCOS susceptibility, with an OR 1.46, 95% CI = 1.007 to 2.14, RR 1.20 (0.99 to 1.45), *p* = 0.049.

### 4.5. Allele Distribution and Genotype Frequency in PCOS Patients and Controls (p-Values) for miR-27a rs895819 A > G Genotypes

In PCOS patients, the genotype frequencies for AA, GA, and GG were 38.09%, 52.38%, and 9.52%, respectively, and in healthy controls, AA, GA, and GG genotype frequencies were 52.17%, 34.78%, and 13.04%, respectively (Table 5). The distribution of miR-27a rs895819 A > G genotypes between PCOS patients and healthy controls was reported to be significant (*p* < 0.031). Furthermore, the frequency of allele G (fG) was observed to be slightly higher among PCOS patients than in healthy controls (0.36 vs. 0.31).

### 4.6. Association between miR-27a rs895819 A > G Genotypes and PCOS Susceptibility as Determined by Multivariate Analysis

As reported in Table 6, our results demonstrate that in the codominant model, the miR-27a rs895819 AG heterozygosity was strongly associated with an enhanced PCOS risk and susceptibility, with OR 2.06, 95% CI = 1.16 to 3.65, RR = 1.42 (1.07 to 1.894), *p* < 0.012. There was a strong association observed between miR-27a AA vs. (GA + GG) genotype in the dominant inheritance model with OR 1.77, 95% CI = 1.035 to 3.034, RR = 1.30 (1.0176 to 1.684), *p* < 0.036. No association was observed in the miR-27a-GG vs. (AA + GA) genotype in the recessive inheritance model, and similarly, it was found that in allelic comparison, the G allele was not associated with PCOS susceptibility with an OR 1.11, 95% CI = 0.742 to 1.6617, RR 1.105 (0.859–1.29), *p*-value = 0.060.

### 4.7. Allele Distribution and Genotype Frequency in PCOS Patients and Controls (p-Values) for miR-423 rs6505162 C > A Genotypes

In PCOS patients, the genotype frequencies for CC, CA, and AA were reported to be 28.57%, 59.04%, and 12.38%, respectively, and in healthy controls, CC, CA, and AA genotype frequencies were 21%, 59%, and 20%, respectively, as shown in Table 7. The distribution of miR-423 rs6505162 C > A genotypes between PCOS patients and healthy controls was not significant (*p* < 0.21). Furthermore, the frequency of allele C (fC) was found to be higher among PCOS patients than in healthy controls (0.58 vs. 0.51), whereas a higher frequency of allele A (fA) was found in healthy controls in comparison with in the PCOS patients (0.49 vs. 0.42).

### 4.8. Association between miR-423 rs6505162 C > A Genotypes and PCOS Susceptibility as Determined by Multivariate Analysis

As reported in Table 8, our results demonstrated that in the codominant model, the miRNA-423 –CC genotype vs. CA and AA genotypes was not associated with an enhanced PCOS susceptibility, with OR 0.73, 95% CI = 0.37 to 1.42, RR = 0.84 (0.580 to 1.229), *p* < 0.36. Further, in the dominant inheritance model, miR-423-CC genotype vs. miR-423-(CA + AA) genotype was also not associated with increased PCOS susceptibility, with OR 0.66, 95% CI = 0.35 to 1.26, RR = 0.80 (0.580 to 1.153), *p* < 0.21. In addition, as shown in the table, in allelic comparison, the C allele was not associated with PCOS susceptibility, with OR 0.85, 95% CI = 0.49–1.08, RR 0.85 (0.702–1.04), *p*-value = 0.12.

### 4.9. Allele Distribution and Genotype Frequency in PCOS Patients and Controls (p-Values) for miR-196a-2 rs11614913 C > T Genotypes

The distribution of miR-196a-2 rs11614913 C > T genotypes between PCOS patients and healthy controls was reported to be significant, as shown in Table 9 (*p* < 0.021). In PCOS patients, the genotype frequencies CC, CT, and TT were 42.60%%, 47.82%, and 9.56%, respectively, and in healthy controls, CC, CT, and TT genotype frequencies were 60.86%, 33.04%, and 6.08%, respectively. Furthermore, the frequency of allele T (fT) was found to be significantly higher among PCOS cases than in healthy controls (0.23 vs. 0.17), whereas the frequency of allele C (fC) was reported to be higher among healthy controls than in the PCOS patients (0.83 vs. 0.77).

### 4.10. Association between miR-196a-2 rs11614913 C > T Genotypes and PCOS Susceptibility as Determined by Multivariate Analysis

As reported in Table 10, our results demonstrated that in the codominant model, the miR-196a-2 rs11614913-CT genotype was strongly associated with enhanced PCOS susceptibility, with OR 2.06, 95% CI = 1.191 to 3.58, RR = 1.43 (1.080 to 1.918), *p* < 0.009. There was a strong association observed between miR-196a-2 CC genotype vs. miR-196a-2-(CT + TT) genotype in a dominant inheritance model with OR 2.09, 95% CI = 1.238 to 3.546, RR = 1.45 (1.106 to 1.902), *p* < 0.005, that was associated with increased PCOS susceptibility. Further, no association was reported between miR-196a-TT genotype vs. miR-196a-2 (CC + CT) genotype in the recessive inheritance model. The results also show that in the case of allelic comparison, the T allele was strongly associated with PCOS susceptibility, with an OR 1.72, 95% CI = 1.140 to 2.603, RR 1.33 (1.057–1.68), *p*-value = 0.009.

## 5. Discussion

PCOS exhibits clinical and symptomatic characteristics that encompass a multitude of atypical effects in various metabolic and cellular signaling pathways, which makes it a complex disease [31]. The disease shows highly heterogeneous manifestations with regard to the reproductive health, with infertility, hirsutism, and hyperandrogenism; metabolic conditions that include T2DM, insulin resistance, and cardiovascular diseases; and psychological distress, including depression and anxiety, which seriously affect the quality of life [32]. Recent recommendations from the panel of experts and consensus resolutions have identified PCOS as a key health concern among women and have emphasized the need for evidence-based initiatives for appropriate modification of diagnostic criteria and treatment strategies for an enhanced well-being of the patients [33,34]. Our study showed marked alterations in the metabolic markers in patients with PCOS as compared to the healthy control. The majority of the patients suffered from T2DM, with high serum glucose, HbAc and altered insulin profile with insulin resistance. There is a four-fold enhanced risk of T2DM associated with PCOS, and it has been estimated that the population attributable risk is 19–28%, which is avoidable with the strategic management of PCOS in young women [35]. The patients’ altered BMI and lipid profiles were also related to the adverse association of PCOS with obesity. Previously, it has been reported that abdominal obesity and hyperandrogenism are linked to the dyslipidaemia in PCOS, and higher serum testosterone has been shown to be adversely associated with insulin levels and HOMA IR in such patients [36]. Due to the complexity of metabolic alterations in PCOS, one of the common observations in these patients includes dyslipidemia, with notably high serum LDL and lower HDL levels [37]. It has been proposed that such an altered composition of HDL is more prevalent in obese PCOS patients (BMI > 27), while in the case of lean patients, the serum HDL levels remain largely unaffected [38]. Thus, there may be variations in these serum parameters in PCOS when the degree of obesity and BMI are taken into consideration [39]. The endocrine impairment in patients show altered levels of FSH, LH, and progesterone, and substantial androgen excess. Studies have shown that one of the key features of PCOS is debilitated gonadal steroid hormone negative feedback to the brain’s GnRH neuronal network, which regulates fertility [40]. It is believed that such neuroendocrine impairment is associated with androgen excess and consequent reproductive dysfunction. It is noted that studies have provided evidence that link genetic variations with the risk and susceptibility towards complex diseases [41]. A polymorphism in coding sequence expresses itself with the altered level of protein phenotype in cases of a non-synonymous change, whereas in non-coding regions, the polymorphisms are regulatory and relatively difficult to resolve in terms of their effect on gene expression. However, studies mapping the allele-specific effects in determining the risk to a disease trait have been significant for a more personalized approach for prevention and treatment [42]. Even a subtle difference in the activity of the two alleles might be associated with a genetic predisposition to a disease [43]. Several genetic variants of the KLF14 gene on chromosome 7 have been reported to be associated with metabolic diseases such as obesity, T2DM, insulin resistance, and cardiovascular disorders, with a gender bias, showing stronger association in females as compared to males [44,45]. Since PCOS also manifests an altered state of metabolic syndrome with a dominant characteristic of T2DM in advanced stages and represents a female-specific disorder, our study aimed to explore its significance in PCOS. As mentioned, it is to be noted that these metabolic alterations in metabolic syndrome are also the ones that characterize PCOS. Moreover, the prevalence rate of metabolic syndrome has shown an upward trend that is increasing with the cases of obesity worldwide, and the prevalence is significantly higher in women compared with men [46]. KLF14 has been classified as a member of group 3 Krüpple-like factors and acts as a transcription activator [47]. A significant number of genetic variants that are associated with type-2 DM and metabolic disorders are localized 3–48 kb upstream of KLF14, and an association of several SNPs affecting the KLF14 expression levels have been identified in adipose tissues [48,49], making it a master regulator implicated in metabolic syndrome [46]. A meta-analysis examining the effect of rs972283 polymorphism in KLF14 for T2DM investigated five studies with 50,552 cases and 106,535 controls and demonstrated high odd ratios for the risk allele G that was found to be associated with an increased risk of T2DM in a global population [50]. An association of the KLF14 rs4731702 SNP and serum lipids as a predictor for cardiovascular disease has also been reported [51]. We report the strong association of KLF14 rs972283 A > G polymorphism with PCOS. Studies have found that the majority of women with PCOS are either overweight or obese (38–88%), and both the conditions (PCOS and obesity) are interwoven in a complex manner that makes the pathogenesis of such metabolic disorders quite difficult to resolve [52]. Since KLF14 is typically linked to the regulation of gene expressions in adipose tissues, the association of its polymorphic variations with obesity or PCOS might be of significance in disease pathology. MicroRNAs, which are short non-coding RNAs, have emerged as an important regulator of gene expression in the mammalian genome [53]. These miRNAs bind to the 3′UTR of the target mRNA and induce translational repression. There are a number of miRNAs, which serve as key regulators of lipid and glucose homeostasis and insulin signaling, thereby actively participating in metabolic dysregulation [54]. MiRNA-related polymorphisms have been associated with the risk of cardiovascular disease, which potentially contribute to the ethnic disparities observed in the associated risk factors in CVD [55]. In a recent study, certain miRNAs that are involved in the regulation of insulin signaling have been found to have a role in the pathogenesis of T2DM through mechanisms that include interactions of SNP–SNP and SNP–environmental factors [56]. It was reported that miRNA-27a rs895819 A/G polymorphism was found to increase the risk of recurrent spontaneous abortion (RSA), and the non-AA genotypes displayed 2.7 times higher risk of RSA in comparison with the AA genotype, providing a link to its role in female fertility [57]. Our study reports that in the codominant model, the miR-27a rs895819 AG heterozygosity is strongly associated with increased PCOS susceptibility, and a strong association between the miR-27a AA genotype and the miR-27a (GA + GG) genotype was observed in the dominant inheritance model. MiR-196a2 rs11614913 polymorphism has been associated with the risk of coronary artery disease, with a higher risk observed in females and elderly patients > 63 years of age, with certain allelic forms in combination with other SNPs found to be associated with the disease pathogenesis [58]. In a recent meta-analysis with 10 cohort studies, it was reported that the pooled risk of adverse cardiovascular events such as myocardial infarction and ischemic heart disease are higher in PCOS patients [59]. The MiR-196a-2 rs11614913-CT genotype is reported to be strongly associated with increased PCOS susceptibility in our study, and in terms of allelic comparison, it was the T allele that was strongly associated with risk of the disease. A recent study investigating the association of certain microRNAs in endometrosis has demonstrated an association of miR-27a (rs895819) and miR-423 (rs6505162) gene variants with the risk and severity of the disease [60]. The statistical analysis in our study for miR-423 gene polymorphism did not show association of the polymorphic forms with the PCOS disease in all inheritance models tested for the genotype’s allelic frequencies. In summary, our study found polymorphic variations in transcription activator KLF14 and miRNAs 27a and 196a genes as functional polymorphisms, which are associated with the risk and susceptibility of PCOS and might contribute to the disease pathogenesis in the studied population. Moreover, such studies are being increasingly appreciated for a personalized approach in the management of a disease, wherein polymorphic gene variations might contribute to the disease risk, progression, or variations in therapeutic outcomes [61].

## 6. Conclusions

Serum biomarkers, both biochemical and endocrine, including lipids (LDL, HDL, cholesterol), T2DM markers (free insulin, fasting glucose, HOMA-IR), and hormones (LH, FSH, testosterone, and progesterone), showed altered states in PCOS patients. The genotype distribution of KLF14, miR-27a, and miR-196a-2 between PCOS patients and healthy controls were strongly associated with the risk and susceptibility to PCOS, as indicated by statistical significance (*p* < 0.05), except for miR-423 genotypes, which were not found to be associated with PCOS (*p* > 0.05). Similarly, allele A of KLF-14, and T allele of miR-196a2, were strongly associated with the PCOS susceptibility. These results are significant with regard to the identification of certain functional polymorphisms in PCOS. However, future studies with larger sample sizes and in different populations are warranted.

## Figures and Tables

**Table 1 jpm-12-00586-t001:** ARMS-PCR primers for KLF-14 rs972283, miR-27 rs895819, miR-423-rs6505162, and miR-196a2-rs11614913.

Direction	Sequence	Product Size	Temp.
ARMS primers of KLF14 rs972283 genotyping			
KLF14 F1	5′-GTCATAGGTCAAACAGCTAGATATTGGGT-3′	437 bp	60 °C
KLF14R1	5′-TCTACAGGACCAACTCAAATTATGAGGT-3′		
KLF14 F2 (G allele)	5′-TCATTGTATACTTGGAAAAAATCCTACATG-3′	274 bp	
KLF14 R2 (A allele)	5′-TATGTAAAAATAAGTATGCGCCATGCCT-3′	221 bp	
ARMS primers of miR-27a rs895819 genotyping			
miR27a F1	5′-GGCTTGACCCCTGTTCCTGCTGAACT-3′	353 bp	63 °C
miR27a R1	5′-TTGCTTCCTGTCACAAATCACATTGCCA-3′		
miR27a F2 (G allele)	5′-GGAACTTAGCCACTGTGAACACGACTTTGC-3′	184 bp	
miR27a R2 (A allele)	5′-CTTAGCTGCTTGTGAGCAGGGTCCCCA-3′	226 bp	
ARMS primers of miR-423 rs6505162 genotyping			
miR-423 F1	5′-TTTTCCCGGATGGAAGCCCGAAGTTTGA-3′	336 bp	62 °C
miR-423 R1	5′-TTTTGCGGCAACGTATACCCCAATTTCC-3′		
miR-423 F2 (A allele)	5′-TGAGGCCCCTCAGTCTTGCTTCCCAA-3′	228 bp	
miR-423 R2 (C allele)	5′-CAAGCGGGGAGAAACTCAAGCGCGAGG-3′	160 bp	
ARMS primers of Hsa-miR-196a2 rs11614913 genotyping			
miR-196a2 F1	5-ACCCCCTTCCCTTCTCCTCCAGATAGAT-3	297 bp	61 °C
miR-196a2 R1	5-AAAGCAGGGTTCTCCAGACTTGTTCTGC-3		
miR-196a2 F2 (T allele)	5-AGTTTTGAACTCGGCAACAAGAAACGGT-3	199 bp	
miR-196a2 R2 (C allele)	5-GACGAAAACCGACTGATGTAACTCCGG-3	153 bp	

**Table 2 jpm-12-00586-t002:** Comparative biochemical profiling of patients and controls.

Characteristic	Controls ^a^	Cases ^a^	*p* ^b^
Age	28.32 ± 4.12	27.59 ± 4.93	0.226
FBG	5.52 ± 0.87	7.45 ± 2.03	<0.001
Free insulin (mU/mL) ^c^	7.49 ± 2.58	13.18 ± 2.98	<0.001
HbA1c	5.20 ± 0.48	5.35 ± 0.42	0.117
HOMA	1.90 ± 0.89	5.34 ± 4.24	<0.001
TAGs (mmol/L) ^c^	1.63 ± 0.46	1.81 ± 0.62	0.059
Cholesterol (mmol/L) ^c^	1.78 ± 0.59	1.88 ± 0.59	<0.001
LDL (mmol/L) ^c^	1.74 ± 0.57	3.70 ± 1.45	<0.001
HDL (mmol/L) ^c^	1.44 ± 0.27	1.60 ± 0.467	<0.001
Estradiol levels (pmol/L) ^d^	241.23 (138.12–478.76)	247.45 (156.21–502.76)	0.251
FSH levels (mIU/mL) ^d^	1.3 (1.24–1.96)	5.0 (3.96–4.85)	<0.001
LH levels (mIU/mL) ^d^	0.09 (0.09–1.69)	3.42 (0.89–9.86)	<0.001
Testosterone levels (ng/dL) ^d^	23.8 (22.33–20.77)	52.0 (33.99–73.73)	<0.001
Progesterone levels (ng/mL) ^d^	16.52 (2.14–19.24)	20.18 (2.78–36.29)	<0.006
Prolactin levels (µg/L) ^d^	11.45 (6.17–11.64)	15.0 (12.76–16.329)	0.74
BMI (kg/m^2^) ^c^	23.71 ± 2.32	26.2± 2.52	<0.001

^a^ 115 cases and 115 controls. ^b^ Student’s *t*-test for continuous variables (variables with normal distribution), Mann–Whitney *U*-test (variables that were not normally distributed). Values as ^c^ mean ± standard deviation and ^d^ median (interquartile range).

**Table 3 jpm-12-00586-t003:** Association of Krüppel-like factor 14 rs972283 G > A gene variation in PCOS cases and controls.

Subjects	N	GG	GA	AA	Df	X^2^	G	A	*p*-Value
Cases	107	40 (37.38%)	35 (32.71%)	32 (29.90%)	2	8.94	0.54	0.46	0.011
Controls	115	47 (40.86%)	52 (45.21%)	16 (13.91%)			0.64	0.36	

**Table 4 jpm-12-00586-t004:** Association of Krüppel-like factor 14 rs972283 G > A gene variation with PCOS susceptibility.

Genotypes	Healthy Controls	PCOS Cases	OR (95% CI)	Risk Ratio (RR)	*p*-Value
	(*n* = 115)	(*n* = 107)			
Codominant					
KLF14-GG	47	40	(ref.)	(ref.)	
KLF14-GA	52	35	0.79 (0.433–1.442)	0.90 (0.6973–1.171)	0.44
KLF14-AA	16	32	2.35 (1.1286–4.8932)	1.62 (1.0390–2.5280)	0.022
Dominant					
KLF14-GG	47	40	(ref.)	(ref.)	
KLF14-(GA + AA)	68	67	1.15 (0.6747–1.9867)	1.07 (0.8301–1.3856)	0.59
Recessive					
KLF14-(GG + GA)	99	75	(ref.)	(ref.)	
KLF14-AA	16	32	2.64 (1.3496–5.1641)	1.70 (1.1210–2.5990)	0.0046
Allele					
KLF14-G	146	115	(ref.)	(ref.)	
KLF14-A	84	97	1.46 (1.0017–2.1456)	1.20 (0.9968–1.4576)	0.049

**Table 5 jpm-12-00586-t005:** Association of miR-27a rs895819 A > G gene variation in PCOS cases and controls.

Subjects	N	AA	AG	GG	Df	X^2^	A	G	*p*-Value
Cases	105	40 (38.09%)	55 (52.38%)	10 (9.52%)	6.93	2	0.64	0.36	0.031
Controls	115	60 (52.17%)	40 (34.78%)	15 (13.04%)			0.69	0.31	

**Table 6 jpm-12-00586-t006:** Association of miR-27a rs895819 A > G gene variations with PCOS susceptibility.

Genotypes	Healthy Controls	PCOS Cases	OR (95% CI)	Risk Ratio (RR)	*p*-Value
	(*n* = 115)	(*n* = 105)			
Codominant					
miR-27a-AA	60	40	(ref.)	(ref.)	
miR-27a-GA	40	55	2.06 (1.1653–3.650)	1.42 (1.0716–1.894)	0.012
miR-27a-GG	15	10	1.0 (0.4088 to 2.4464)	1.0 (0.6992 to 1.4302)	0.10
Dominant					
miR-27a-AA	60	40	(ref.)	(ref.)	
miR-27a-(GA + GG)	55	65	1.77 (1.035–3.0347)	1.30 (1.0176–1.684)	0.036
Recessive					
miR-27a-(AA + GA)	80	95	(ref.)	(ref.)	
miR-27a-GG	15	10	0.56 (0.2391–1.3183)	0.76 (0.5324–1.0904)	0.185
Allele					
miR-27a-A	140	135	(ref.)	(ref.)	
miR-27a-G	70	75	1.11 (0.742–1.6617)	1.05 (0.8594–1.293)	0.06

**Table 7 jpm-12-00586-t007:** Association of miR-423 rs6505162 C > A gene variation in PCOS cases and controls.

Subjects	N	CC	CA	AA	Df	X^2^	C	A	*p*-Value
Cases	105	30 (28.57%)	62 (59.04%)	13 (12.38%)	2	3.03	0.58	0.42	0.21
Controls	100	21 (21%)	59 (59%)	20 (20%)			0.51	0.49	

**Table 8 jpm-12-00586-t008:** Association of miR-423 rs6505162 C > A gene variation in PCOS cases and controls.

Genotypes	Healthy Controls	PCOS Cases	OR (95% CI)	Risk Ratio (RR)	*p*-Value
	(*n* = 100)	(*n* = 105)			
Codominant					
miRNA-423–CC	21	30	(ref.)	(ref.)	
miRNA-423–CA	59	62	0.73 (0.3795–1.425)	0.84 (0.5801–1.229)	0.36
miRNA-423–AA	20	13	0.45 (0.1862–1.112)	0.67 (0.4428–1.042)	0.08
Dominant					
miR-423–CC	21	30	(ref.)	(ref.)	
miR-423–(CA + AA)	79	75	0.66 (0.3501–1.2615)	0.80 (0.558–1.153)	0.21
Recessive					
miR-423–(CC + CA)	80	92	(ref.)	(ref.)	
miR-423–AA	20	13	0.56 (0.260–1.206)	0.76 (0.5582–1.0551)	0.14
Allele					
miR-423–C	101	122	(ref.)	(ref.)	
miR-423–A	99	88	0.73 (0.4983–1.0867)	0.85 (0.7021–1.0425)	0.123

**Table 9 jpm-12-00586-t009:** Association of miR-196a-2 rs11614913 (C > T) gene variation in PCOS cases and controls.

Subjects	N	CC	CT	TT	Df	X^2^	C	T	*p*-Value
Cases	115	49 (42.60%)	55 (47.82%)	11 (9.56%)	7.7	2	0.77	0.23	0.021
Controls	115	70 (60.86%)	38 (33.04%)	07 (6.08%)			0.83	0.17	

**Table 10 jpm-12-00586-t010:** Association of miR-196a-2 rs11614913 C > T gene variation in PCOS cases and controls.

Genotypes	Controls	PCOS Cases	OR (95% CI)	Risk Ratio (RR)	*p*-Value
	115	115			
Codominant					
miR-196a-2-CC	70	49	(ref.)	(ref.)	
miR-196a-2-CT	38	55	2.06 (1.191 to 3.589)	1.43 (1.080 to 1.918)	0.009
miR-196a-2-TT	07	11	2.24 (0.813 to 6.197)	1.51 (0.831 to 2.751)	0.11
Dominant					
miR-196a-2-CC	70	49	(ref.)	(ref.)	
miR-196a-2-(CT + TT)	45	66	2.09 (1.238 to 3.546)	1.45 (1.106 to 1.902)	0.005
Recessive					
miR-196a-2-(CC + CT)	108	104	(ref.)	(ref.)	
miR-196a-2-TT	07	11	1.63 (0.60 to 4.370)	1.31 (0.723 to 2.372)	0.32
Allele					
miR-196a-2-C	178	153	(ref.)	(ref.)	
miR-196a-2–T	52	77	1.72 (1.140 to 2.603)	1.33 (1.0573 to 1.68)	0.009

## Data Availability

All the associated data for the study has been included in this manuscript.

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
