# Peer review of "Clinical Implications of Krüpple-like Transcription Factor KLF-14 and Certain Micro-RNA (miR-27a, miR-196a2, miR-423) Gene Variations as a Risk Factor in the Genetic Predisposition to PCOS"

_jpm, 2022, doi:10.3390/jpm12040586_

Round 1

Reviewer 1 Report

Comments:

  1. Could the authors please provide evidence of ethical approval for their study please?
  2. Could the authors please clarify how the number of study participants were determined i.e. how was their study powered?
  3. Could the authors please justify the use of parametric statistics for their presentation and analyses please?

Author Response

Comments:

  1. Could the authors please provide evidence of ethical approval for their study please?
  • Research & Research Ethics Committee / Reg. No: H0-07-TU-002/Ethics 10: 0057
  • Date: 16/12/1434
  1. Could the authors please clarify how the number of study participants were determined i.e. how was their study powered?

In the beginning of the study, we had taken help from the Biostastician .He was working as an assistant professor, department of science, University of Tabuk. 

Sample size is a research term used for defining the number of individuals included in a research study to represent a population. The sample size references the total number of respondents included in a study, and the number is often broken down into sub-groups by demographics such as age, gender, and location so that the total sample achieves represents the entire population. Determining the appropriate sample size is one of the most important factors in statistical analysis. Following formula

Sample Size Justification: As per the incidence of PCOS cases in Tabuk region, the study included 115 subjects of PCOS and 115 controls. As per the frequency of cases of PCOS patients registered per week, the target was achieved and even exceeded.

Finite Population Scenario: If the population (N) is known, then follow the Yamane method:

(1)   nY = N / (1 + Ne2)

... where N = known population and e = error level or % percent confidence interval or alpha level. For 0.95 confidence interval, e = 0.05. For example, if the population is 50,000, what is the minimum sample size?

nY = 50,000 / (1 + 50,000(0.0025) ,nY = 50,000 / 126  ,nY = 396.83
Under this method for finite population, the minimum sample size is 396.83 or about 397 counts.

(ii) Non-Finite Population Scenario: A second scenario involves unknown population size, i.e. non-finite population. The following formula is used for non-finite population case:

(2)   nnf = Z2σ2 / E2

... where Z = critical value for Z; σ = estimated population standard deviation; and E = standard error which is given by: E = σ / sqrt(ntest). Under this method, a test sample (ntest) must be taken as an initial sample in order to obtain descriptive statistics and use equations (1) and (2) above to calculate the value of σ. A potential problem under this method is the assumption of normality. it may be a good idea to test the data distribution to verify whether the data is normally distributed. If it is not normally distributed, we might look to other methods for minimum sample determination.

  1. Could the authors please justify the use of parametric statistics for their presentation and analyses please?

A parametric statistical test makes an assumption about the population parameters and the distributions that the data came from. These types of test includes Student's T tests and ANOVA tests, which assume data is from a normal distribution.

Hardy-Weinberg equilibrium was studied by a χ2 test to compare the observed genotype frequencies within the case-control groups. P-value was considered significant when it was < 0.05.

Student T test ,Chi-square analysis and Fisher exact test were performed to compare KLF-14 rs972283, miR-27 rs895819, miR-423-rs6505162 and miR-196a2-rs11614913 genotyping frequency with various pathological aspects.

Multivariate analysis -A multivariate analysis based on logistic regression like odds ratio (OD) and risk ratio (RR) with 95% confidence intervals (CI) were calculated for each group to estimate the association between Krüppel-like Factor 14 rs972283 G>A , miR-27 rs895819, miR-423-rs6505162 and miR-196a2-rs11614913 genotypes and risk to PCOS .

Reviewer 2 Report

The manuscript by Mir R et aimed to study the influence of allelic variations for miRNAs 27a (rs895819 A>G), 196a2 (rs11614913 C>T), 423 (rs6505162C>A) and transcription factor KLF14 (rs972283 A>G) gene polymorphism on the risk and susceptibility to PCOS. There are several critical issues with the manuscript that need to be addressed.

Major

1)The sample size used for the study is very small and hence authors should be careful towards the interpretation of polymorphic variations in miRNA with risk and susceptibility of PCOS.

2) The other major issue is the novelty of the subject and the biomarkers used. miR-27a rs895819 and miR-423 rs6505162 risk association with PCOS is already reported (Jaffer S et al 2022).

3) Although KLF14 as the author mentioned is “typically linked to the regulation of gene expressions in adipose tissues, the association of its polymorphic variations with obesity or PCOS might be of significance in disease pathology”. Multiple studies show that KLF14 expression is higher in females, but the effect of the genetic variants on KLF14 is similar in males and females (Small KS 2018, Civelek M 2017). Authors need to provide sufficient evidence or molecular mechanisms that suggest KLF14 rs4731702 SNP associations with PCOS if any.

4) There is a significant overlap of paragraphs between abstract and introduction. The introduction part needs further improvement and conciseness.

Minor

  • The abstract provided by the authors is not concise and wordy. Highly recommend that the authors follow the journal’s manuscript submission guidelines.
  • The manuscript has many grammatical errors and typos. Highly advised to authors to proofread the manuscript from a native English speaker.
  • Please provide line numbers throughout the manuscript. It would be easier for reviewers to pinpoint exact lines and paragraphs for correction.

Author Response

Major

  • The sample size used for the study is very small and hence authors should be careful towards the interpretation of polymorphic variations in miRNA with risk and susceptibility of PCOS.

We found that the miRNAs 27a (rs895819 A>G), 196a2 (rs11614913 C>T), 423 (rs6505162C>A) frequency among all the study participants is in compliance to the HWE. The genotype distributions and allele frequencies of the SNPs located in the miRNAs 27a (rs895819 A>G), 196a2 (rs11614913 C>T), MIR423 (rs6505162C>A) showed no deviation in HWE (all p > 0.05) (χ2 = 2.24 p 0.13) in the control group Thus, we chose 10% samples from normal control group randomly to review genotyping results, showing that the accuracy rate was more than 99%.

It is understood that a lower sample size is required for testing more common SNPs with stronger effect sizes and increased LD between marker allele and disease allele. A lower sample size is required to study a common disease than a rare disease.

The larger the sample size is the smaller the effect size that can be detected. The reverse is also true; small sample sizes can detect large effect sizes. Thus an appropriate determination of the sample size used in a study is a crucial step in the design of a study.

However, to accommodate the reviewer’s suggestion we have included a line in the conclusion for further verification with large sample size.

The genotype distributions and allele frequencies of the miRNAs 27a (rs895819 A>G showed no deviation in HWE in controls

Genotypes

*Observed #

Expected #

Homozygote reference:

60

55.7

Heterozygote:

40

48.7

Homozygote variant:

15

10.7

Var allele freq:

0.30

115

Chi-squared value =

3.667091837

Chi-squared test P value =

0.055497

(if < 0.05 - not consistent with HWE)

The genotype distributions and allele frequencies of the miR-423 rs6505162 C>A showed no deviation in HWE in controls

Genotypes

*Observed #

Expected #

Homozygote reference:

21

25.5

Heterozygote:

59

50.0

Homozygote variant:

20

24.5

Var allele freq:

0.50

100

Chi-squared value =

3.244249818

Chi-squared test P value =

0.071674

(if < 0.05 - not consistent with HWE)

The genotype distributions and allele frequencies of the miR-196a-2 rs11614913 (C>T)  showed no deviation in HWE in controls

Genotypes

*Observed #

Expected #

Homozygote reference:

70

68.9

Heterozygote:

38

40.2

Homozygote variant:

7

5.9

Var allele freq:

0.23

115

Chi-squared value =

0.357396711

Chi-squared test P value =

0.549956

(if < 0.05 - not consistent with HWE)

  • The other major issue is the novelty of the subject and the biomarkers used. miR-27a rs895819 and miR-423 rs6505162 risk association with PCOS is already reported (Jaffer S et al 2022).

Please note that this work of Jaffer S et.al. is on endometriosis and not on PCOS. However, we have included this citation in the discussion to accommodate reviewer’s concern. As it can be seen that this paper and several others have been published in 2022 and our work is also at par with this timing as the current manuscript was submitted to the journal in January 2022.

3) Although KLF14 as the author mentioned is “typically linked to the regulation of gene expressions in adipose tissues, the association of its polymorphic variations with obesity or PCOS might be of significance in disease pathology”. Multiple studies show that KLF14 expression is higher in females, but the effect of the genetic variants on KLF14 is similar in males and females (Small KS 2018, Civelek M 2017). Authors need to provide sufficient evidence or molecular mechanisms that suggest KLF14 rs4731702 SNP associations with PCOS if any.

Several studies have implicated the transcription factor KLF14, a member of the Krüpple-like factor family (KLF), in the development of metabolic diseases, including obesity, insulin resistance, and T2D (Yang and Civelek, 2020). It is to be noted that these metabolic abnormalities in metabolic syndrome are also the ones that characterize PCOS. Moreover, metabolic syndrome prevalence has been increasing with the rise of obesity worldwide, with significantly higher prevalence in women compared with men (Yang and Civelek, 2020). It is therefore a rational approach to investigate KLF14 gene polymorphism in PCOS, wherein metabolic syndrome is a hallmark characteristic. Considering the reviewers apprehensions we have included this statement and the reference in the discussion.

4) There is a significant overlap of paragraphs between abstract and introduction. The introduction part needs further improvement and conciseness.

Since abstract is a summarized paragraph of the manuscript and needs to be read as a stand-alone piece to get an idea of the manuscript, it is obvious that there will be some overlapping with the actual body of the manuscript. However, considering reviewer’s suggestion we have minimized some overlapping.

Minor

  • The abstract provided by the authors is not concise and wordy. Highly recommend that the authors follow the journal’s manuscript submission guidelines.

We have revised the abstract following the reviewer’s suggestion

  • The manuscript has many grammatical errors and typos. Highly advised to authors to proofread the manuscript from a native English speaker.

     Thanks for pointing out the need for grammatical corrections. The manuscript has been proofread.

  • Please provide line numbers throughout the manuscript. It would be easier for reviewers to pinpoint exact lines and paragraphs for correction.

Done

Reviewer 3 Report

In this study, Mir R et al. demonstrate that the distribution of Krüppel-like Factor 14 (KLF14) rs972283 G>A, miR-27a rs895819 A>G and miR-196a-2 rs11614913 C>T genotypes have a strong association with polycystic ovary syndrome (PCOS). Based on the current data, I have some questions and suggestions below. 1. The order of the table is wrong from table 4, please confirm. 2. As shown in Table 2, the authors stated that BMI was significantly changed in PCOS patients, but the results showed lower BMI values in the PCOS group, how could this be explained? Also, is it contradictory that the concentration of Serum HDL, a scavenger of lipid metabolism, was significantly increased in the PCOS group? 3. Why are KLF14, miRNA 27a, miRNA 196a2, and miRNA 423 specifically targeted to investigate the effects of their allelic variants and gene polymorphisms on PCOS risk and susceptibility? Please add an explanation to the discussion paragraph. 4. The abstract needs to be simplified. I suggest that the abstract and the conclusion at the end of the article should have different depictions to avoid repeated statements. 5. The subscripts "c" and "d" in Table 2 are missing in the notes, please add them below the table.

Author Response

  1. The order of the table is wrong from table 4, please confirm.

Thank you for pointing out. It is now corrected.

  1. As shown in Table 2, the authors stated that BMI was significantly changed in PCOS patients, but the results showed lower BMI values in the PCOS group, how could this be explained? Also, is it contradictory that the concentration of Serum HDL, a scavenger of lipid metabolism, was significantly increased in the PCOS group?

The was an error in reporting the BMI which has now been corrected. Thanks for this valuable comment. We have explained the HDL issue as raised by the reviewer in the discussion which was indeed required for the clarity.

“Due to the complexity of metabolic alterations in PCOS, one of the common observations in these patients includes dyslipidemia, with notably high serum LDL and lower HDL levels (Kim and Choi 2013). It has been proposed that such an altered composition of HDL is more prevalent in obese PCOS patients (BMI>27), while in case of lean patients the serum HDL levels remains largely unaffected (Rajkhowa et.al. 1997). Thus, there may be variations in these serum parameters in PCOS when the degree of obesity and BMI are taken into consideration (Holte et.al. 1994)”

  1. Why are KLF14, miRNA 27a, miRNA 196a2, and miRNA 423 specifically targeted to investigate the effects of their allelic variants and gene polymorphisms on PCOS risk and susceptibility? Please add an explanation to the discussion paragraph.

In the light of the reviewer’s suggestion we have justified the rational for all of these polymorphic gene variations in the discussion (Please see the text in red font)

  1. The abstract needs to be simplified. I suggest that the abstract and the conclusion at the end of the article should have different depictions to avoid repeated statements. 5. The subscripts "c" and "d" in Table 2 are missing in the notes; please add them below the table.

 Done

Round 2

Reviewer 1 Report

The authors have satisfactorily responded to my comments.

Reviewer 2 Report

Accept in present form

Reviewer 3 Report

I think the author has given a full response to the questions raised in the first peer review, so it can be accepted.